# Widely targeted metabolomics reveals differences in metabolites of *Paeonia lactiflora* cultivars

Yonghui Li[1]*, Yingying Tian[2], Xiaojun Zhou[1], Xiangmeng Guo[1], Huiyuan Ya[3]*, Shipeng Li[1], Xiangli Yu[1], Congying Yuan[1], Kai Gao[4]

1 School of Life Sciences, Luoyang Normal University, Luoyang, Henan, China, 2 School of Life Sciences, Shaanxi Normal University, Xi'an, Shanxi, China, 3 School of Food and Drug, Luoyang Normal University, Luoyang, Henan, China, 4 Luoyang Academy of Agricultural and Forestry Sciences, Luoyang, Henan, China

* huiyongli8209@126.com (YL); 25460678@qq.com (HY)

**Data Availability Statement:** All relevant data are within the manuscript and its Supporting information files.

**Funding:** This work was supported by the Project of General Project of National Natural Science

## Abstract

### Introduction

*Paeonia lactiflora* contains diverse active constituents and exhibits various pharmacological activities. However, only partial identification of biologically active substances from *P. lactiflora* has been achieved using low-throughput techniques. Here, the roots of *P. lactiflora*, namely, Fenyunu (CK), Dafugui (DFG), and Red Charm (HSML), were studied. The primary and secondary metabolites were investigated using ultrahigh-performance liquid chromatography-electrospray ionization-tandem mass spectrometry (UPLC-ESIMS/MS).

### Methods

The chemical compounds and categories were detected using broadly targeted UPLC–MS/MS. Principal component analysis (PCA), orthogonal partial least-squares discriminant analysis (OPLS-DA), and hierarchical clustering analysis (HCA) were carried out for metabolites of different varieties of *P. lactiflora*.

### Results

A total of 1237 compounds were detected and classified into 11 categories. HCA, PCA, and OPLS-DA of these metabolites indicated that each variety of *P. lactiflora* was clearly separated from the other groups. Differential accumulated metabolite analysis revealed that the three *P. lactiflora* varieties contained 116 differentially activated metabolites (DAMs) involved in flavonoid, flavone, and flavonol metabolism. KEGG pathway analysis revealed that, in 65 pathways, 336 differentially abundant metabolites (DMs) were enriched in the CK and DFG groups; moreover, the type and content of terpenoids were greater in the CK group than in the DFG group. The CK and HSML groups contained 457 DMs enriched in 61 pathways; the type and amount of flavonoids, terpenoids, and tannins were greater in the CK group than in the HSML group. The DFG and HSML groups contained 497 DMs enriched in 65 pathways; terpenoids and alkaloids were more abundant in the HSML variety than in the DFG variety.

Foundation of China (Grant Number: 31870697) and the Natural Science Foundation of Henan Province (Grant Number: 202300410287). The funders had no role in study design, data collection and analysis, decision to publish, or preparation of the manuscript.

**Competing interests:** The authors have declared that no competing interests exist.

## Conclusions

A total of 1237 compounds were detected, and the results revealed significant differences among the three *P. lactiflora* varieties. Among the three *P. lactiflora* varieties, phenolic acids and flavonoids composed the largest and most diverse category of metabolites, and their contents varied greatly. Therefore, CK is suitable for medicinal plant varieties, and DFG and HSML are suitable for ornamental plant varieties. Twelve proanthocyanidin metabolites likely determined the differences in color among the three varieties.

## Introduction

*Paeonia lactiflora* is a perennial herb from the *Paeonia* L. of the Paeoniaceae family. It is also known as "Jiangli," "Licao," and "Yurong" [1]. *P. lactiflora* has a long history of cultivation, which can be traced back to the Xia and Shang Dynasties. *P. lactiflora* is mainly found in the northern and northeast regions, as well as in Shaanxi and Gansu provinces in northwest China. Sichuan, Zhejiang, Shandong and Anhui provinces also have large areas of cultivation. It is also widely distributed in Korea, Japan, Mongolia, Siberia, and Russia [2]. It is a famous ornamental plant and a popular traditional Chinese medicine [3]. According to Sheng Nong's herbal classic, it is used "to expel pathogenic qi, relieve abdominal pain, eliminate blood impediment, dispel hard accumulation, regulate chills and fever, ease the pain, promote urination, and replenish qi" [4]. The roots of *P. lactiflora* exhibit promising anticancer, anti-inflammatory, antioxidative, and liver-protecting effects [5–8]. Monoterpenoid glycosides, such as paeoniflorin and albiflorin, are the main medicinal components of *P. lactiflora* [9, 10]. Miao Yanping et al. [11] also isolated oxypaeoniflorin, galloyl paeoniflorin, benzoylpaeoniflorin, etc., from *P. lactiflora*.

The 'Fenyunu' variety is a representative medicinal variety in China; it uses peony roots to produce traditional Chinese medicinal herbs such as white peony and red peony, and it is referred to as CK in this study. Two other varieties, Dafugui (DFG) and Red Charm (HSML), were used as the experimental subjects. The 'Dafugui' variety (DFG) is widely cultivated in domestic gardens as a representative ornamental plant. 'Red Charm' (HSML) is an imported variety from Europe that is classified as a hybrid peony variety group [12] and combined with Officinalis × Lactiflora, with the female parent being *Paeonia officinalis* from the subsect. Paeonia, and the male parent was *Paeonia lactiflora* Pall. from the subsect. Albiflorae. In this study, both 'Fenyunu' and 'Dafugui' belonged to *Paeonia lactiflora* Pall. HSML is an ornamental, soft-stemmed hybrid of *P. officinalis* and *P. lactiflora* and is imported from Europe. The Fenyunu and Dafugui varieties used in the study are classified as *P. lactiflora*. The Red Charm cultivar has both similarities and differences with *P. lactiflora*.

Metabolomics based on UPLC–MS/MS is a quick, highly accurate, and highly sensitive approach for the detection of metabolites [13–16]. Widely targeted metabolomic analysis combines the benefits of untargeted and targeted metabolomics. The use of multiple reaction monitoring (MRM) adds novelty to the technique, as does its qualitative and quantitative accuracy, high sensitivity, high throughput, and broad coverage [17, 18]. Several research groups in China have analyzed differentially abundant metabolites using widely targeted metabolomics. Differences in diverse metabolite samples affect their taste [19], color [20], nutritional composition [21], medicinal components, etc. The active ingredients of *P. lactiflora* and their preparations have been analyzed using HPLC [22], UPLC–MS [23, 24], UPLC-Q-TOF-MS [25], HPLC-ESI-MS [26], and other methods in China and abroad. Widely targeted metabolomics

with UPLC–MS/MS for analyzing the metabolites of different *P. lactiflora* varieties, such as 7-methoxycoumarin, oxindole, protocatechualdehyde, acetryptine, naringenin-4',7-dimethyl ether, etc., has not been reported. The current study is the first report in which a broadly targeted metabolomic approach was used to identify the chemical compositions and differential accumulation metabolites (DAMs) among the CK, DFG, and HSML varieties of *P. lactiflora*. This paper sheds light on the metabolic pathways involved in *P. lactiflora* and provides a scientific foundation for its application.

## Materials and methods

### Plant materials

All *P. lactiflora* samples were sourced from "The National Peony and Peony Germplasm Bank" (In September 2020, National Forestry and Grass Leaf Administration Announces List, http:// chinahhxh.com/indexs.html) at Luoyang Academy of Agriculture and Forestry Sciences, Luoyang, Henan, China, in mid-December 2021. Disease-free or insect-free peony roots with a diameter greater than 2.5 cm were selected. Located by Gao Kai, Director of the Germplasm Resources Center and identified by Professor Jin Mingxian from the Plant Classification Teaching and Research Office of Luoyang Normal University, these plants are classified as three varieties: "Fenyunu" (CK), "Dafugui" (DFG), and "Red Charm" (HSML). The *P. lactiflora* varieties Fenyunu (CK), Dafugui (DFG), and Red Charm (HSML) were used (S1 Table in S1 File, Fig 1). Fenyunu's flowers are light pink with a single petal, while Dafugui's flowers are red and resemble roses, featuring few petaloid stamens inserted between the petals. Red Charm's flowers are deep red and have a crown-like appearance (Fig 1). The plant samples were freeze-dried with liquid nitrogen and stored at –80˚C. Three biological replicates were performed for each variety for independent analysis. Five tuberous roots from five mature *P. lactiflora* plants were collected and thoroughly mixed into each replicate.

### Sample extractions

The samples were freeze-dried under vacuum in a freeze-drying machine (Scientz-100F). Freeze-dried vacuum samples were pulverized using a laboratory mixing grinder (Germany, Retsch GmbH) for 1.5 min at 30 Hz. Chemicals extracted from 70% methanol can cover most of the metabolites in plants, and the extraction efficiency is greater. Therefore, the crushed powder (0.1 g) was dissolved in methanol (1.2 mL, 70%) and extracted via the vortex extraction method at 4˚C overnight. After centrifugation at $10,000 \times g$ for 10 min, the extraction liquids were subsequently filtered through a Millipore filter (0.22 μm pore size, SCAA-104; ANPEL, Shanghai, China) for LC–MS analysis. UPLC–MS analysis was performed on a QTRAP 6500 triple quadrupole-linear ion trap mass spectrometer (AB Sciex, Framingham, USA). The analytical conditions used were detailed (HPLC column system, Waters AC).

### UPLC–MS/MS analysis

The analysis was conducted using the following instrument systems. A UPLC–ESI–MS/MS system was used (UPLC, SHIMADZU Nexera X2; MS, Applied Biosystems 4500 Q-TRAP, www.appliedbio-systems.com.cn/). These two systems were used for data acquisition. The analysis conditions were as follows: HPLC columns: Agilent SB-C18 (2.1 mm × 100 mm, 1.8 μm); solvent system: solvent A: ultrapure water with 0.1% formic acid, and solvent B: acetonitrile with 0.1% formic acid. In the gradient program, at 0 min, the B phase proportion was 5%; within 9 min, the B phase proportion increased linearly to 95% and was maintained at 95% for 1 min; 10–11 min, the B phase proportion decreased to 5% and was balanced at 5% to

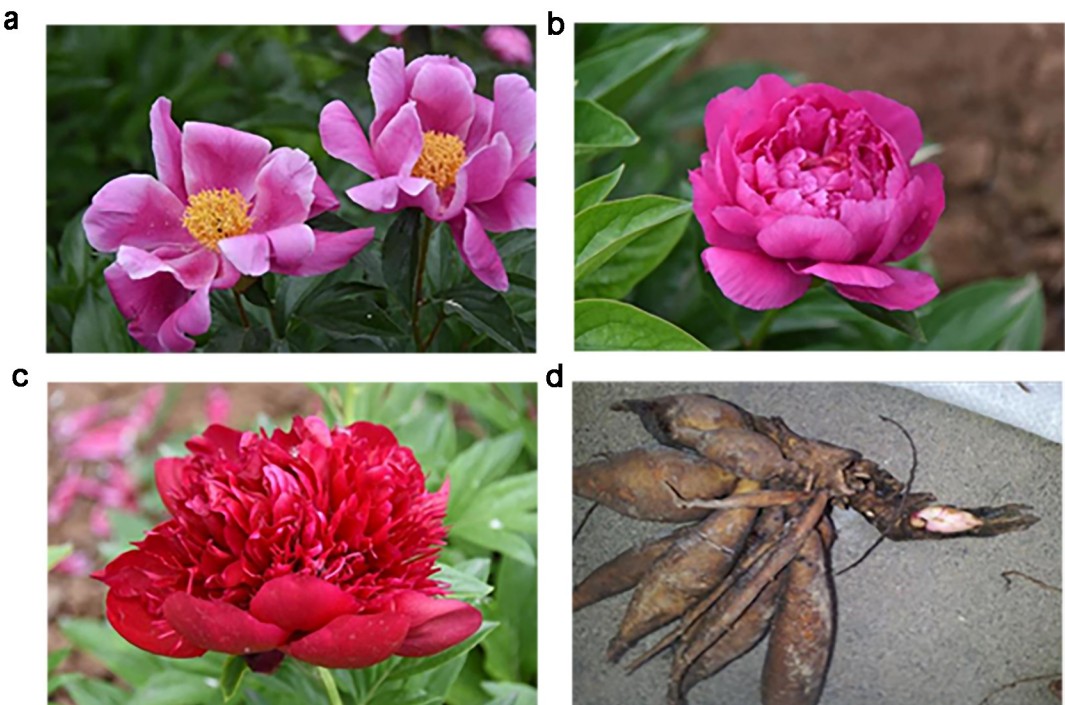

**Fig 1. Three varieties of *Paeonia lactiflora* Pall.** (a) Fenyunu; (b) Dafugui; (c) Red Charm; (d) *Paeonia lactiflora* Pall's root.

14 min; and the flow rate was 0.35 mL/min. The outflows were connected to an electrospray ionization Q-TRAP-MS system alternatively. The analysis conditions followed those of Chu et al. [27].

### ESI-Q TRAP-MS/MS

Triple quadrupole (QQQ) scans and linear ion trap (LIT) analysis were conducted on a triple Q-TRAP 6500 UPLC–MS/MS system. Both the positive and negative modes were operational and processed using Analyst 1.6.3 (Toronto, Canada). The ESI source operating parameters included the following: turbojet was the ion source, the heat source temperature was 550˚C, the negative ion mode was –4500 V, and the positive ion mode was +5500 V. A higher collisional gas parameter (CAD, collision activated dissociation) was maintained. Instrument calibration was implemented using polypropylene glycols (10 µmol/L and 100 µmol/L) in LIT and QQQ patterns, respectively. Multireaction monitoring (MRM) was adopted to acquire QQQ scans with nitrogen gas as the collision medium. Similar experimental procedures to those of Zhang et al. [28] were successfully employed. Wide target quantitative mode: The quantitative gold standard (MRM) mode and characteristic ion quantitative mode were used; the Q3, RT, DP, and CE of each substance were optimized, and the peak shape was improved. Accuracy: Quantitative analysis was performed using an AB Sciex 6500 QTRAP. The MRM mode can eliminate nontarget ion interference, improving the accuracy and reproducibility of the quantification.

### Metabolite qualitative and quantitative analysis

**Qualitative analysis.** The mixed samples were qualitatively analyzed via high-resolution mass spectrometry AB Sciex TripleTOF6600. Mass spectrometry data were analyzed using the

MWDB self-built database (MetWare Biotechnology Co., Ltd., Wuhan, China) [29]. Public metabolite databases, such as MassBank (http://www.massbank.jp/), KNAPSAcK (http://kanaya.naist.jp/KNApSAcK/), and ChemBank (http://chembank.med.harvard.edu/compounds), were utilized for structural analysis of the metabolites.

**Quantitative analysis.** Quantitation was performed using an AB Sciex 6500 QTRAP. To eliminate interference from nontarget compounds with different molecular weights in the MRM pattern, only the precursor ions of the target material were selected from the triple quadrupole. The mass spectra peak areas of identical metabolites from different samples were integrated. These peaks were subsequently corrected using MultiQuant software [30]. The material content data, parent ions, and characteristic fragment ions used are detailed in S1 Data. The data in the numerical table in the S1 Data table were analyzed via the scientific counting method; the relative content of metabolites, without units, was determined by calculating the peak area formed by the characteristic ions of each substance in the detector. Q1: Molecular weight of parent ions after the material was added to the electrospray ionization source. Q3: Characteristic fragment ions. Sample column: Sample relative content.

## Metabolite data analysis

The metabolite analytical data of the three peony varieties were analyzed using Analyst 1.6.3 software for statistical analyses. After unit variance scaling, principal component analysis (PCA) and hierarchical clustering analysis (HCA) were performed to assess sample groupings [31]. PCA was performed with the built-in statistical package in R software (www.rproject.org/), and the prcomp function parameter scale was set to true to normalize the data using unit variance scaling (UV). Metabolite content data were normalized using Unit Variance Scaling, and a heatmap was drawn using the R software ComplexHetmap package to perform HCA analysis on the accumulation patterns of metabolites among different samples. Additionally, orthogonal partial least squares-discriminant analysis (OPLS-DA) and the supervised multivariate method maximized the metabolomic differences between paired samples. The relative significance of metabolites in each generation of the OPLS-DA model was evaluated using variable importance in the projection (VIP). OPLS-DA involved performing a log2 transformation on the raw data, followed by mean centering, and then using the OPLSR Anal function of the MetaboAnalystR package of R software for analysis. Differentially abundant metabolites (DMs) were screened using differential multiple log2 processing, VIP, and [VIP + fold change (FC)] double screening. A volcano map was drawn with ggplot2 (V3.3.0) in R (version 3.5.1). A volcano plot was generated to show the difference in the relative content of metabolites between the two varieties and the statistical significance of the difference. Venn diagrams were drawn using the Venn diagram (V1.6.20) package in R (V3.5.1). A Venn diagram was constructed to show the common DMs of flavonoids among the CK, DFG, and HSML varieties. KEGG enrichment: The process of annotating metabolites to the KEGG pathway was performed according to the cpd_ID of differentially abundant metabolites, the p value of each pathway was calculated by a hypergeometric test, and the ratio of the number of differentially expressed metabolites in the corresponding pathway to the total number of metabolites detected and annotated by this pathway was expressed by the Rich factor. DMs were annotated to the Kyoto Encyclopedia of Genes and Genomes (KEGG) pathways using the KEGG database to study the metabolic network [32] (Kanehisa & Goto, 2000).

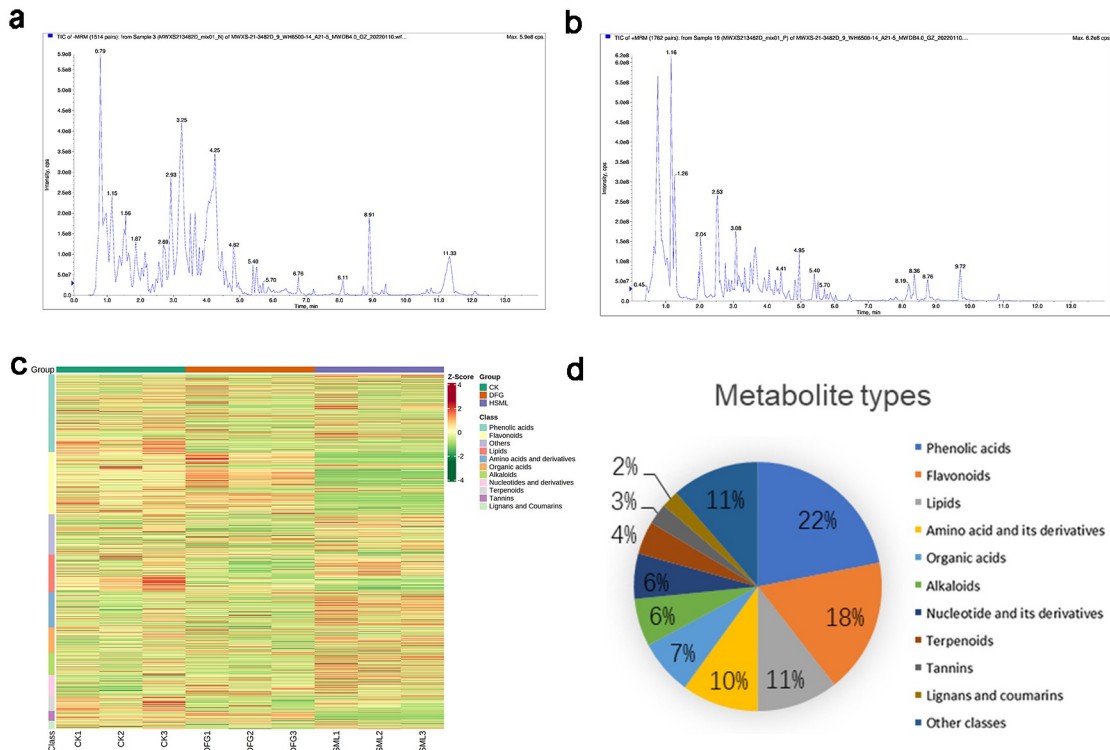

**Fig 2. Qualitative and quantitative analysis of metabolites in *P. lactiflora*.** Total ion current diagram of the mass spectra in (a) negative ion mode; (b) positive ion mode; (c) heatmap of the metabolites from the roots of *P. lactiflora*; and (d) types and proportions of the identified metabolites from *P. lactiflora*.

## Results

### *P. lactiflora* metabolics profiling

The instrument demonstrated satisfactory signal stability, as evidenced by the overlapping total ion current (TIC) curves of the metabolites, indicating that the residence time matched the peak intensity (Fig 2a and 2b). A total of 1237 metabolites spanning 11 categories were identified. These categories included 271 phenolic acids, 217 flavonoids, 131 lipids, 123 amino acids and derivatives, 89 organic acids, 77 alkaloids, 73 nucleotides and their derivatives, 53 terpenoids, 34 tannins, 28 lignans and coumarins, and 141 others. Among these, phenolic acids (22%), flavonoids (18%), amino acids and their derivatives (10%), and lipids (11%) represented the four primary metabolite types (Fig 2c and 2d).

### Multivariate statistical analysis

Multivariate statistical analysis was used to analyze three or more variables and further classify all metabolites identified from the CK, DFG, and HSML varieties using PCA and OPLS-DA.

**HCA and PCA.** HCA categorized the CK, DFG, and HSML varieties into three groups, highlighting different accumulation patterns among the three variables (Fig 3a). A PCA score map (Fig 3b) revealed differences between the three varieties by revealing a separation trend between the different groups [33]. The quality control sample originated from the mix. The two principal components, PC1 and PC2, in the PCA score plot explained 41.55% and 25.27%, respectively, of the variability. Minor differences were observed between the three replicates of

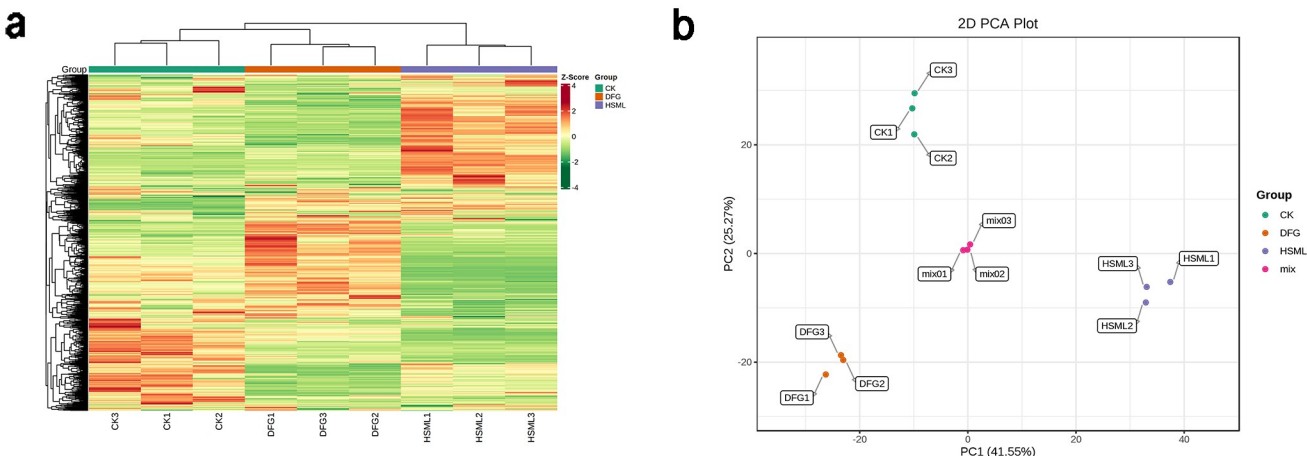

**Fig 3. Metabolomic data analysis of *P. lactiflora* roots.** (a) Heatmap clustering of the identified metabolites of the CK, DFG, and HSML varieties; (b) Principal component analysis (PCA) of the CK, DFG, and HSML varieties.

each *P. lactiflora* variety, with the three biological replicates of each variety clustered in specific areas. This clear separation indicated significant differences in metabolites among the Fenyunu, Dafugui, and Red Charm varieties.

**OPLS-DA.** PCA may not identify variables with lower correlations, but OPLS-DA overcomes this limitation. OPLS-DA, a supervised pattern recognition method, showed significant differences in distribution between the samples in each group on the left and right sides of the confidence scale. This confirmed the significant differences among the Fenyunu, Dafugui, and Red Charm varieties (Fig 4a–4c).

## Differentially abundant metabolites in the roots of *P. lactiflora*

Volcano map. Multiple changes in the quantitative information of metabolites were compared among the CK and DFG, CK and HSML, and DFG and HSML groups. The data from each group were processed by the difference multiple log2, and the top 20 differentially abundant metabolites are listed in Table 1. The variations in proanthocyanidin expression with flower color regulation are shown in S2 Table in S1 File. The DMs were further filtered by combining VIP values ≥1 and fold change ≥2 or ≤0.5. The different volcano maps were compared (Fig 5). A total of (i) 336 compounds (139 upregulated, 197 downregulated metabolites) were

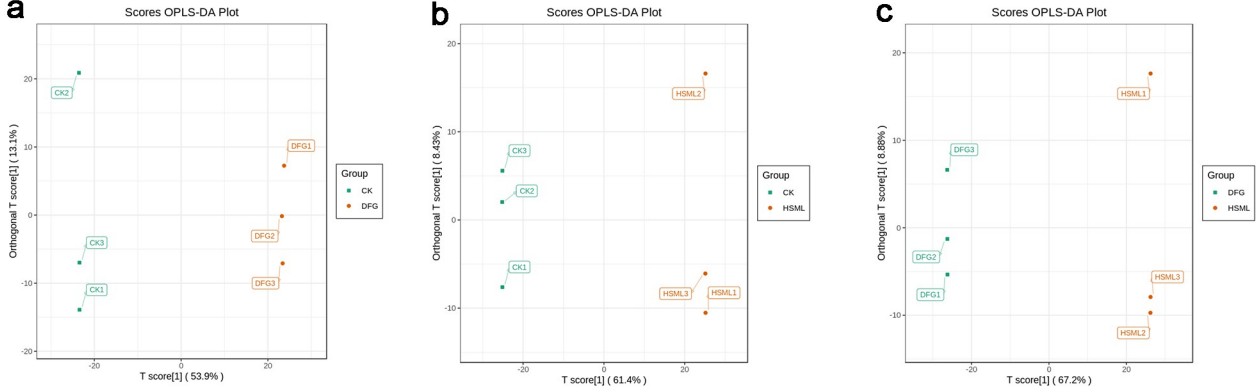

**Fig 4. OPLS-DA score diagrams.** (a) CK and DFG score diagram; (b) CK and HSML score diagram; (c) DFG and HSML score diagram.

**Table 1. The top 20 metabolites after multiple differential treatments.**

| Group | Classification | Compound | up/down regulated |
|---|---|---|---|
| CK vs. DFG1 | Flavonoids | Luteolin-7-O-neohesperidoside (Lonicerin) | up |
| CK vs. DFG2 | Alkaloids | Oxindole | up |
| CK vs. DFG3 | Lignans and Coumarins | 7-Methoxycoumarin | up |
| CK vs. DFG4 | Alkaloids | Naringenin-4',7-dimethyl ether | up |
| CK vs. DFG5 | Phenolic acids | 2-Methoxy-4-methylphenol | up |
| CK vs. DFG6 | Flavonoids | Kaempferide (3,5,7-Trihydroxy-4'-methoxyflavone)* | up |
| CK vs. DFG7 | Phenolic acids | 3-Hydroxy-5-Methylphenol-1-O-Glucoside | up |
| CK vs. DFG8 | Others | D-Fucose | up |
| CK vs. DFG9 | Phenolic acids | 4-O-Sinapoylquinic acid | up |
| CK vs. DFG10 | Alkaloids | 5-Hydroxyquinoline | up |
| CK vs. DFG10 | Lipids | LysoPC 16:1 | down |
| CK vs. DFG9 | Nucleotides and derivatives | Uridine 5'-diphosphate | down |
| CK vs. DFG8 | Terpenoids | "6'-O-benzoyl-4""-hydroxy-3""-methoxy-paeoniflorin" | down |
| CK vs. DFG7 | Alkaloids | Cimicifugamide A | down |
| CK vs. DFG6 | Phenolic acids | 3,4,5-Tri-O-Galloylshikimic acid | down |
| CK vs. DFG5 | Phenolic acids | Benzaldehyde | down |
| CK vs. DFG4 | Others | N-(beta-D-Glucosyl) nicotinate | down |
| CK vs. DFG3 | Phenolic acids | 4-O-Glucosyl-4-hydroxybenzoic acid | down |
| CK vs. DFG2 | Terpenoids | 6'-O-benzoylalbiflorin | down |
| CK vs. DFG1 | Terpenoids | paeonisothujone | down |
| CK vs. HSML1 | Flavonoids | 6,7,8,3',4'-Pentamethoxyflavanone | up |
| CK vs. HSML2 | Alkaloids | Oxindole | up |
| CK vs. HSML3 | Phenolic acids | Sinapaldehyde-4-O-Glucoside | up |
| CK vs. HSML4 | Phenolic acids | 2-Methoxy-4-methylphenol | up |
| CK vs. HSML5 | Lignans and Coumarins | 7-Methoxycoumarin | up |
| CK vs. HSML6 | Flavonoids | Quercetin-3-O-(2"-O-Xylosyl) rutinoside | up |
| CK vs. HSML7 | Alkaloids | Acetryptine | up |
| CK vs. HSML8 | Alkaloids | 4-Coumaroylcholine | up |
| CK vs. HSML9 | Phenolic acids | 3,4-Dimethylbenzoic acid | up |
| CK vs. HSML10 | Others | D-Fucose | up |
| CK vs. HSML10 | Phenolic acids | 2,6-Dimethoxybenzoic acid | down |
| CK vs. HSML9 | Flavonoids | Orientin-2"-O-galactoside | down |
| CK vs. HSML8 | Flavonoids | Pinocembrin-7-O-glucoside (Pinocembroside) | down |
| CK vs. HSML7 | Others | 2',4'-Dihydroxy-6'-methoxyacetophenone | down |
| CK vs. HSML6 | Phenolic acids | 2',4',6'-Trihydroxyacetophenone | down |
| CK vs. HSML5 | Amino acids and derivatives | Trans-4-Hydroxy-L-proline | down |
| CK vs. HSML4 | Flavonoids | 3,4,2',4',6'-Pentahydroxychalcone-4'-O-glucoside | down |
| CK vs. HSML3 | Terpenoids | 6'-O-benzoylalbiflorin | down |
| CK vs. HSML2 | Phenolic acids | 1-O-Vanilloyl-D-Glucose | down |
| CK vs. HSML1 | Phenolic acids | Mucic acid-1,4-lactone-3,5-di-O-gallate | down |
| DFG vs. HSML1 | Phenolic acids | Protocatechualdehyde | up |
| DFG vs. HSML2 | Flavonoids | 6,7,8,3',4'-Pentamethoxyflavanone | up |
| DFG vs. HSML3 | Phenolic acids | Benzaldehyde | up |
| DFG vs. HSML4 | Phenolic acids | 3,4,5-Tri-O-Galloylshikimic acid | up |
| DFG vs. HSML5 | Phenolic acids | Homovanillic acid; 4-Hydroxy-3-methoxyphenylacetic acid | up |
| DFG vs. HSML6 | Phenolic acids | 4-O-Glucosyl-4-hydroxybenzoic acid | up |
| DFG vs. HSML7 | Phenolic acids | Sinapaldehyde-4-O-Glucoside | up |

(*Continued*)

**Table 1.** (Continued)

| Group | Classification | Compound | up/down regulated |
|---|---|---|---|
| DFG vs. HSML8 | Alkaloids | L-Tyramine | up |
| DFG vs. HSML9 | Nucleotides and derivatives | Uridine 5'-diphosphate | up |
| DFG vs. HSML10 | Lipids | LysoPC 16:1 | up |
| DFG vs. HSML10 | Flavonoids | 6-C-Methylquercetin-3-O-rhamnoside | down |
| DFG vs. HSML9 | Phenolic acids | 3,4-Dihydroxyacetophenone | down |
| DFG vs. HSML8 | Others | 2',4'-Dihydroxy-6'-methoxyacetophenone | down |
| DFG vs. HSML7 | Phenolic acids | 2,5-dihydroxyphenylethanone | down |
| DFG vs. HSML6 | Tannins | 2α,3α-Epoxy-5,7,3',4'-tetrahydroxyflavan-(4β-8-epicatechin)* | down |
| DFG vs. HSML5 | Tannins | 2α,3α-Epoxy-5,7,3',4'-tetrahydroxyflavan-(4β-8-catechin)* | down |
| DFG vs. HSML4 | Amino acids and derivatives | Trans-4-Hydroxy-L-proline | down |
| DFG vs. HSML3 | Flavonoids | 3,4,2',4',6'-Pentahydroxychalcone-4'-O-glucoside | down |
| DFG vs. HSML3 | Phenolic acids | 2',4',6'-Trihydroxyacetophenone | down |
| DFG vs. HSML1 | Phenolic acids | 1-O-Vanilloyl-D-Glucose | down |

differentially expressed (DMs) between the CK and DFG varieties; (ii) 457 compounds (149 upregulated, 308 downregulated) were differentially expressed between the CK and HSML varieties; and (iii) 497 compounds (223 upregulated, 274 downregulated) were differentially expressed between the DFG and HSML varieties (Fig 5).

Heatmap clustering. The heatmap clustering illustrates significant differences among the DMs within the three *P. lactiflora* varieties (Fig 6). The DMs accumulated in varying proportions in these three varieties. There were significant differences in the type and relative content of flavonoids and phenolic acids between the CK and DFG groups. The CK variety contained a greater variety and greater amount of terpenoids than the DFG variety. When considering the CK and HSML categories, CK had higher levels of flavonoids, terpenoids, and tannins, whereas HSML had lower levels. In contrast, other constituents accumulated in significantly different amounts. Phenolic acids had distinct types and contents, with flavonoids and tannins contributing more to DFG than to HSML. Moreover, HSML had relatively more terpenoids, alkaloids, and other substances than did DFG.

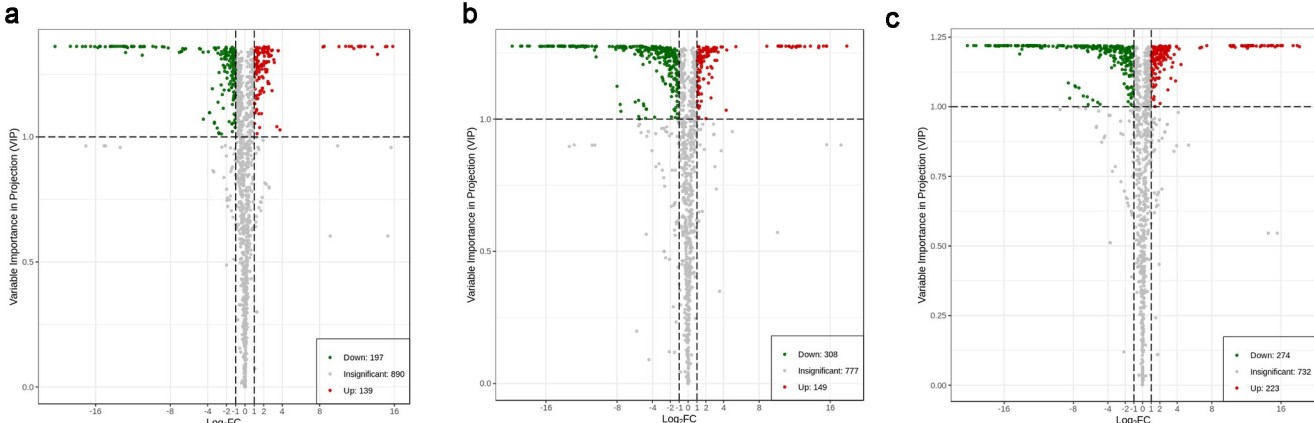

**Fig 5. Volcano maps of the differentially abundant metabolites.** (a) CK vs. DFG group; (b) CK vs. HSML group; (c) DFG vs. HSML group. Note: Ordinate: VIP values; Abscissa: log2 |FC|; red dots: upregulated metabolites; green dots: downregulated metabolites.

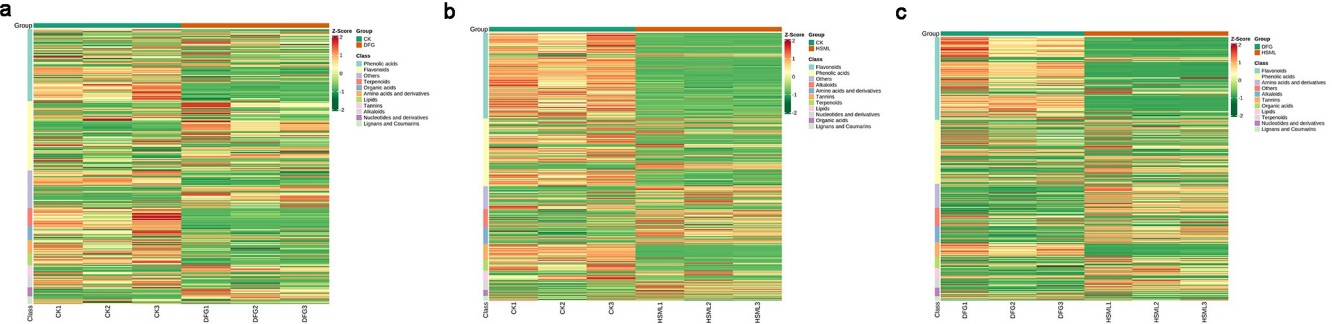

**Fig 6. Heatmap of the differentially abundant metabolites.** (a) CK vs. DFG group; (b) CK vs. HSML group; (c) DFG vs. HSML group. Note: Abscissa: sample name; Ordinate: differentially abundant metabolites. Different colors in the heatmap represent the normalized values for the relative content of the differentially abundant metabolites; red dots represent upregulated metabolites; and green dots represent downregulated metabolites.

A Venn diagram was used to visualize the number of common and endemic substances between the different comparison groups. The Venn plot illustrates 116 common differentially abundant metabolites, including 49 flavonoids, 29 phenolic acids, 10 tannins, 10 other substances, 8 terpenoids, 6 alkaloids, 3 amino acids and their derivatives, 1 nucleotide derivative, and 1 coumarin (Fig 7).

## Differentially abundant metabolite KEGG pathways

The KEGG database enables researchers to study genes, expression information, and metabolite content as a whole network. Differentially abundant metabolites in the CK and DFG, CK and HSML, and DFG and HSML groups were annotated in the KEGG database to extract corresponding pathway information. A total of 336 different metabolites involved in 65 pathways

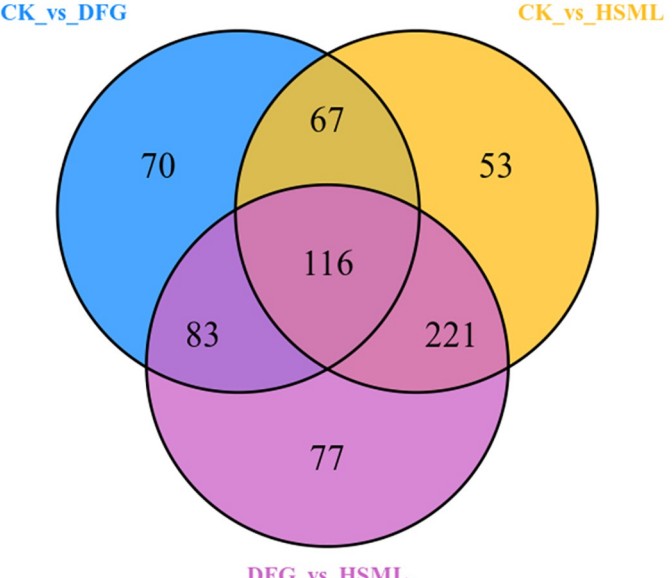

**Fig 7. Venn plot of group differences.**

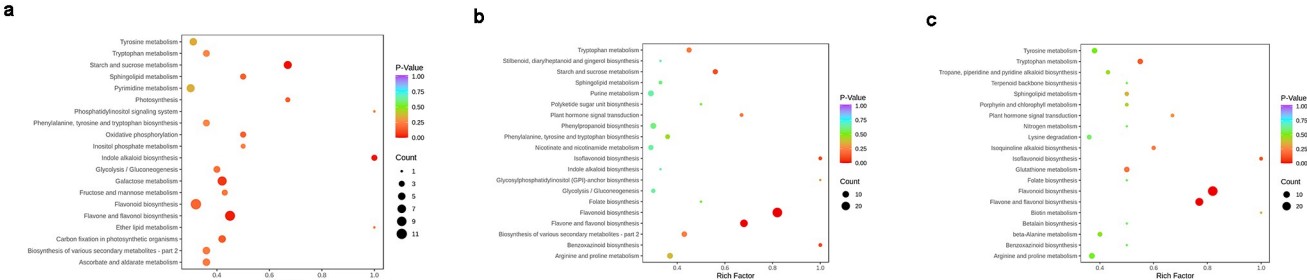

**Fig 8. KEGG pathway of the differential metabolites.** (a) CK vs. DFG group; (b) CK vs. HSML group; (c) DFG vs. HSML group. Note: Abscissa: rich factor; Ordinate: path name. The dot size represents the number of enriched differentially abundant metabolites, and the dot color represents the p value; a redder dot represents a more significant enrichment.

were enriched in the CK and DFG groups, 457 metabolite groups involved in 61 pathways were enriched in the CK and HSML groups, and 497 metabolites were associated with 65 pathways in the DFG and HSML groups (Fig 8). The metabolites enriched in the CK and DFG groups were mainly associated with the biological synthesis of flavonoids, flavones and flavonols, as well as metabolic pathways such as galactose, sucrose, and starch metabolism, among others. The different metabolites enriched in the CK and HSML groups correlated with flavonoid synthesis, flavone and flavonol synthesis, and sucrose and starch metabolism. The different metabolites enriched in the DFG and HSML groups were primarily linked to flavonoid, flavone, and flavonol synthesis and tryptophan metabolism. The flavonoid biosynthesis pathway and the flavone and flavonol biosynthesis pathways overlapped in all these comparisons.

## Discussion

*P. lactiflora* is known for its medicinal and ornamental value and contains various bioactive substances, such as flavonoids, terpenoids, lipids, and phenolic compounds [8–10, 14, 17, 20, 34]. This study identified more than 1237 metabolites belonging to 11 categories in *P. lactiflora*. Flavonoids, phenolic acids, lipids, amino acids, their derivatives, alkaloids, organic acids, nucleotides, their derivatives, and terpenoids were the major metabolite types (Fig 2), significantly enriching our understanding of the chemical constituents of *P. lactiflora*.

UPLC–MS/MS and widely targeted metabolomics analyses revealed differentially abundant metabolites in three representative *P. lactiflora* varieties. Correlation coefficients exceeding 0.8 between all pairs of samples indicated significant correlations. Although the differences within each *P. lactiflora* variety group were minimal, significant distinctions were observed between the three groups (Figs 4 and 5). A total of 1237 differentially expressed metabolites spanning 11 categories were detected among the three *P. lactiflora* varieties.

Current research on the medicinal components of *P. lactiflora* has focused primarily on compounds such as total glucosides of peony (TGP) and phenols [5, 35, 36]. TGP contains monoterpenoids and their glycosides, contributing to immunomodulation, anti-inflammation, and liver protection. Conversely, polyphenols scavenge oxygen free radicals and prevent oxidative damage [37–39]. In the present study, TGP was found to be composed of terpenoids, especially sesquiterpenoids such as paeoniflorin, oxidized paeoniflorin, and galactyl paeoniflorin, as well as other active components. Phenolic substances predominantly included phenolic acids and flavonoids. These compounds possess high medicinal potential.

The DM clustering heatmap HCA revealed a greater quantity and variety of terpenoids in the CK and DFG groups (Fig 6). The CK group displayed greater amounts of flavonoids,

terpenoids, and tannins than did the HSML group. The terpenoid content, known for its pharmacodynamic properties, was speculated to be greater in the Fenyulu variety than in the other varieties, suggesting that this variety is suitable for medicinal purposes. In contrast, the Dafugui variety is ideal for ornamental purposes. This conclusion was consistent with Kang's research in 2011 [40]. Furthermore, red charm is suggested to be a primary hybrid variety for ornamental purposes due to its distinctive flower shape, unique color, and robust stems, making it an important choice for fresh-cut flowers. The significant difference in flavonoid content between the HSML, CK, and DFG varieties may be attributed to their geographical origins and variations in temperature and humidity. This observation aligns with the research findings of Li Jian and Si Shiying [41, 42].

Venn diagram analysis revealed 116 common differentially abundant metabolites, including 49 DMs of flavonoids, 29 phenolic acids, 10 tannins, 10 other substances, 8 terpenoids, 6 alkaloids, 3 amino acids and their derivatives, 1 nucleotide derivative, and 1 coumarin (Fig 7). KEGG database annotation revealed significant differences in flavonoid, flavone, and flavonol metabolic pathways among the CK and DFG, CK and HSML, and DFG and HSML varieties (Fig 8). Flavonoids play a key role in synthesizing flower pigments [43] and exhibit various health benefits, including antibacterial, antioxidant, antitumor, and analgesic properties, as well as treatment of conditions such as liver injury, heart rhythm, and disorders [44–46]. Previous studies have reported more than 20 types of flavonoids, including dihydro apigenin, catechins, anthocyanins, and catechins, in *P. lactiflora* [47, 48] (Heim *et al.*, 2002; Wu *et al.*, 2016). This study identified 217 species of flavonoids. Since plant anthocyanins are primarily flavonoids and carotenoids [49] (Mori *et al.*, 2009), the differentially accumulated metabolites enriched in flavonoid, flavone, and flavonol biosynthesis could be responsible for the color variations among the three *P. lactiflora* varieties. Among the 12 proanthocyanidin metabolites, the contents of 2α,3α-epoxy-5,7,3',4'-tetrahydroxyflavan-(4β-8-epicatechin/catechin)*, procyanidin A2 B3, C1 and C2 in the three *P. lactiflora* varieties were DFG > CK > HSML (S2 Table in S1 File). The difference in color among the three *P. lactiflora* varieties may be related to the content of proanthocyanidin.

## Conclusions

This study marks the first application of a broadly targeted metabolomics approach to analyze the chemical components of the Fenyunu, Dafugui, and Red Charm varieties of *P. lactiflora*. A total of 1237 presumed metabolites were identified in the roots of *P. lactiflora*. Flavonoids and phenolic acids were the predominant components and exhibited significant variations among the different varieties. Most of the differentially expressed metabolites in the varieties were associated with the biosynthesis pathways of flavonoids, flavones, and flavonols. The Fenyunu variety was characterized by its dominance of terpenoids, especially paeoniflorin, along with phenolic acids and flavonoids. In contrast, the Dafugui variety lacked terpenoids and other bioactive compounds. Red Charm peonies, which are primarily ornamental, contain several terpenoids and other medicinal components. In summary, this study serves as a valuable reference for the utilization, research, breeding, and genetic enhancement of valuable *P. lactiflora* varieties with potential applications in medicinal plants and functional foods.

## Supporting information

**S1 File.**
(DOCX)

**S2 File.**
(RAR)

**S1 Data.**
(XLSX)

**S2 Data.**
(XLSX)

**S3 Data.**
(XLSX)

**S4 Data.**
(XLSX)

## Acknowledgments

The authors sincerely thank Professor Huiping Ma, Luoyang Academy of Agricultural and Forestry Sciences, Luoyang, Henan, for providing helpful suggestions and advice.

## Author Contributions

**Conceptualization:** Xiangli Yu.

**Data curation:** Yonghui Li, Yingying Tian, Shipeng Li, Congying Yuan.

**Formal analysis:** Yonghui Li, Xiangmeng Guo, Shipeng Li, Xiangli Yu.

**Funding acquisition:** Xiaojun Zhou, Huiyuan Ya.

**Investigation:** Yingying Tian.

**Methodology:** Yonghui Li, Xiaojun Zhou, Huiyuan Ya, Shipeng Li, Congying Yuan.

**Project administration:** Yonghui Li, Huiyuan Ya.

**Resources:** Xiaojun Zhou, Huiyuan Ya, Kai Gao.

**Software:** Huiyuan Ya, Xiangli Yu.

**Validation:** Xiangmeng Guo, Huiyuan Ya, Shipeng Li.

**Visualization:** Yonghui Li, Xiangmeng Guo.

**Writing – original draft:** Yingying Tian.

**Writing – review & editing:** Yonghui Li, Xiangmeng Guo, Congying Yuan, Kai Gao.

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
