## [Decision Letter · Decision Letter 0]

26 Oct 2023

PONE-D-23-30697Widely targeted metabolomics reveals differences in metabolites of Paeonia lactiflora cultivarsPLOS ONE

Dear Dr. Li,

Thank you for submitting your manuscript to PLOS ONE. After careful consideration, we feel that it has merit but does not fully meet PLOS ONE’s publication criteria as it currently stands. Therefore, we invite you to submit a revised version of the manuscript that addresses the points raised during the review process.

ACADEMIC EDITOR PLOS requires an ORCID iD for the corresponding author in Editorial Manager on papers submitted after December 6th, 2016. Please ensure that you have an ORCID iD and that it is validated in Editorial Manager. To do this, go to ‘Update my Information’ (in the upper left-hand corner of the main menu), and click on the Fetch/Validate link next to the ORCID field. This will take you to the ORCID site and allow you to create a new iD or authenticate a pre-existing iD in Editorial Manager. Please see the following video for instructions on linking an ORCID iD to your Editorial Manager account: https://www.youtube.com/watch?v=_xcclfuvtxQ Rechecking the cover letter is necessary. It has a different manuscript title.

We look forward to receiving your revised manuscript.

Kind regards,

Sairah Hafeez Kamran, PhD

Academic Editor

PLOS ONE

Journal Requirements:``

Reviewers' comments:

Reviewer's Responses to Questions

**Comments to the Author**

1. Is the manuscript technically sound, and do the data support the conclusions?

Reviewer #1: Yes

Reviewer #2: Partly

2. Has the statistical analysis been performed appropriately and rigorously? 

Reviewer #1: N/A

Reviewer #2: No

3. Have the authors made all data underlying the findings in their manuscript fully available?

Reviewer #1: Yes

Reviewer #2: No

4. Is the manuscript presented in an intelligible fashion and written in standard English?

Reviewer #1: Yes

Reviewer #2: Yes

5. Review Comments to the Author

Reviewer #1: Review Report:

The manuscript entitled “Widely targeted metabolomics reveals differences in metabolites of Paeonia lactiflora cultivars” seems to be an interesting study in which authors made attempts to contribute the field.

Note: I reviewed the file named (PONE-D-23-30697) downloaded from journal online portal and all my comments are according to the line numberings of the pdf file “PONE-D-23-30697”

Widely targeted metabolomics reveal differences in metabolites of Paeonia lactiflora cultivars.

Detailed Comments/Recommendations:

Title:

Title of the study is quite justifiable.

Abstract:

Lines 17-19: Objective of the study is not clear

In line 17 and 18 provide full forms of CK, DFG and HSML.

Line 20-23: Results are mentioned in this portion instead of methodological procedures. Please write down method and procedure here.

Results are poorly expressed.

Conclusion in abstract must be comprehensive.

Key Words: Authors used 3 out of 4 same words as mentioned in title. The key word should be different from the words used in title so that the reader may explore the study with different more focused words.

Introduction:

Kindly add the introduction (family, genus, distribution, cultivation etc) of P. lactiflora.

In line 63 add on the names of some metabolites of P. lactiflora.

Materials and Methods:

Plant Materials:

From where plant material was identified? Add on the details like identification or herbarium no etc.

Plant extraction:

Line 80: Kindly mention the centrifuging in gram units instead of rpm.

Reference is missing for plant extraction method at end of line 83.

Results & Discussion:

Discussion is well written by the authors.

Conclusion:

Satisfactory.

References:

The authors are advised to review all the references for the strict adherence of the uniformity of Journal’s references style.

Lines 284-285: Reference is incomplete. Add page number.

Lines 332-334: Correct the reference according to the journal format.

Concluding Remarks:

The paper needs above mentioned corrections along with the English Language proof readings by some language expert. Then after fulfilling the corrections the paper will be acceptable.

Reviewer #2: Comments:

The authors study the differences in metabolites of Paeonia lactiflora cultivars which shows metabolomic analysis of the different cultivars. It is well written, but very brief and following comments should be addressed.

Major concerns:

1. Pure metabolomics lacks standards. It is recommended to confirm the major predicted components by classical HPLC quantification with the standards.

2. Quantification of the components was not shown. Also, the identification confirmation based on a mother total ion and two distinct fragments was not shown.

3. How did authors confirm the species varieties? Are there official names and genetic identification available, could authors perform a genetic comparison study?

4. How did the authors decide on the HPLC condition of up to only 25% B solvent. In this way, most of the nonpolar compounds will not be eluted.

5. Which part of plant did the authors extract? Sometimes deep freezing may change the tissues rapidly, affecting the constituents

6. How did authors select 70% methanol as the solvent? Why methanol and not ethanol? Was the condition of the sample kept equal to preserve the proanthocyanin contents?

7. Please explain aim and methodology of volcano map, Venn plot, KEGG.

8. How the statistical analysis was performed?

9. Supplementary information is not clear.

Please check the reference for previous work

doi: 10.1016/j.jep.2020.113281

6. PLOS authors have the option to publish the peer review history of their article (what does this mean?). If published, this will include your full peer review and any attached files.

Reviewer #1: **Yes: **Dr. Aamir Mushtaq

Reviewer #2: **Yes: **Michal Korinek

---

## [Author Response · Author response to Decision Letter 0]

23 Nov 2023

Response to Reviewers [PONE-D-23-30697]

Dear Sir/Madam：

Thank you very much for your last letter and advice. This letter is in response to two reviewers questions.

Reviewer 1：

Abstract:

1.Lines 17-19: Objective of the study is not clear

Reply: Have revised in line 18-20. Such as:“Here, the roots of P. lactiflora, namely Fenyunu (CK), Dafugui (DFG), and Red Charm (HSML), were studied. The primary and secondary metabolites were investigated using ultrahigh-performance liquid chromatography-electrospray ionization-tandem mass spectrometry (UPLC-ESIMS/MS).”

2.In line 17 and 18 provide full forms of CK, DFG and HSML.

Reply: Have already provided in line 18-19. Such as: Fenyunu (CK), Dafugui (DFG), and Red Charm (HSML).

3. Line 20-23: Results are mentioned in this portion instead of methodological procedures. Please write down method and procedure here.

Reply: Have revised in line 21-23.

4.Results are poorly expressed.

Reply: Have revised in line 24-32.

5. Conclusion in abstract must be comprehensive.

Reply: Have revised in line 33-37.

Key Words: Authors used 3 out of 4 same words as mentioned in title. The key word should be different from the words used in title so that the reader may explore the study with different more focused words. 

Reply: Have revised in line 39-40.

Introduction:

1.Kindly add the introduction (family, genus, distribution, cultivation etc) of P. lactiflora.

Reply: Have revised in line 42-47.

2. In line 63 add on the names of some metabolites of P. lactiflora.

Reply: Such as, the names of some metabolites of P. lactiflora have 7-Methoxycoumarin, Oxindole, Protocatechualdehyde, Acetryptine, Naringenin-4',7-dimethyl ether, etc in line 76-77.

Materials and Methods:

Plant Materials:

From where plant material was identified? Add on the details like identification or herbarium no etc.

Reply: Have revised in line 83-89.

Plant extraction:

Line 80: Kindly mention the centrifuging in gram units instead of rpm.

Reply: Has been changed to 10,000 g in line 99.

Reference is missing for plant extraction method at end of line 83.

 Reply: Plant extraction method was vortex extraction method in line 98-99.

References:

The authors are advised to review all the references for the strict adherence of the uniformity of Journal’s references style.

Reply: Reviewed and revised.

Lines 284-285: Reference is incomplete. Add page number.

Reply: Page numbers 47 have been added and formatting corrected in line 321.

Lines 332-334: Correct the reference according to the journal format.

Reply: Have corrected the reference according to the journal format in line 365-367.

Reviewer #2: Comments:

The authors study the differences in metabolites of Paeonia lactiflora cultivars which shows metabolomic analysis of the different cultivars. It is well written, but very brief and following comments should be addressed.

Major concerns:

 1. Pure metabolomics lacks standards. It is recommended to confirm the major predicted components by classical HPLC quantification with the standards.

Reply: (1) High resolution mass spectrometry AB sciex TripleTOF6600 was used for qualitative detection of mixed samples for plant wide target, and then AB sciex6500 QTRAP was used for quantitative determination. Combined with the advantages of non-targeted and targeted metabolomics, high resolution mass spectrometry was used for accurate characterization. Triple quadrupole mass spectrometry with high sensitivity, high specificity and excellent quantitative capability is used as a complementary tool. 

(2) The matching parameters in metabolite identification are Q1 precise molecular mass, secondary fragment, retention time, isotope. 

(3) Qualitative and quantitative mass spectrometry analysis of metabolites in samples is based on the MetWare database (MWDB) and Multiple Reaction Monitoring (MRM). The identification of metabolites is based on the exact mass of metabolites, MS2 fragments, MS2 fragment isotope distribution and retention time (RT). Through the intelligent secondary spectrum matching method independently developed by MYWAY Company, the secondary spectrum and RT of metabolites in the project samples are intelligentially matched with the secondary spectrum and RT of the company database. The MS and MS2 tolerances are set to 20 ppm and the RT tolerances to 0.2min.

（4）The broad target detection method is to build a database of the standard product, and the metabolites without the standard product will be compared with the secondary spectrum in the public database or the literature. The source of substance identification at the time of database construction can be marked.

（5）Wide target quantitative mode: quantitative gold standard MRM mode, characteristic ion quantitative; Q3, RT, DP, CE of each substance will be optimized, and the peak shape will be better. Accuracy: AB sciex6500 QTRAP quantification; In the MRM model, the four-pole first screens the precursor ions (parent ions) of the target substance, and excludes the ions corresponding to other molecular weight substances to preliminarily eliminate the interference. After ionization induced by the collision chamber, the precursor ion breaks to form a lot of fragment ions, and then a characteristic fragment ion is selected through the triple four-pole filtration to eliminate the interference of non-target ions, so that the quantification is more accurate and the repeatability is better. After the metabolic mass spectrometry data of different samples were obtained, the peak area integral was performed for all the mass spectral peaks of substances, and the mass spectral peaks of the same metabolite in different samples were integral corrected (Fraga et al. 2010).

Fraga, C.G., et al., Signature-discovery approach for sample matching of a nerve-agent precursor using liquid chromatogd chemometrics. Anal Chem, 2010. 82(10): p. 4165-73.

Wide target methodology literature: Chen, W, Gong, L Guo, Z, et al,A Novel Integrated Method for Large-Scale Detection, Identification, and Quantification of Widely Targeted Metabolites: Application in the Study of Rice Metabolomics. Molecular Plant, 2013, 6(6): 1769-1780.

The relevant literature with high scores has been published：Zhao ML, Ren YJ , Wei W, Yang JM, Zhong QW and Zheng L. Metabolite Analysis of Jerusalem Artichoke(Helianthus tuberosus L.) Seedlings in Response to PolyethyleneGlycol-Simulated Drought Stress, Int.J. Mol. Sci. 2021, 22, 3294.

 2. Quantification of the components was not shown. Also, the identification confirmation based on a mother total ion and two distinct fragments was not shown. 

Reply: Material content data, parent ions, and characteristic fragment ions are detailed in “Supporting Information2”: “ALL_sample_data.xlsx“. The data in the numerical table in the ALL_sample_data table uses the scientific counting method and is the relative content of metabolites, without units, by calculating the peak area formed by the characteristic ions of each substance in the detector; among Q1: Molecular weight of parent ions after the material is added by the electrospray ion source; Q3: Characteristic fragment ions; Sample column: Sample relative content (In lines 132-137).

3. How did authors confirm the species varieties? Are there official names and genetic

identification available, could authors perform a genetic comparison study?

Reply: All the P. lactiflora samples were sourced from "The National Peony and Peony Germplasm Bank" (In September 2020, National Forestry and Grass Leaf Administration Announces List, http://chinahhxh.com/indexs.html ) at Luoyang Academy of Agriculture and Forestry Sciences. Provided by Gao Kai, Director of the Germplasm Resources Center, and identified by Professor Jin Mingxian from the Plant Classification Teaching and Research Office of Luoyang Normal University, they are classified as three varieties: "Fenyunu" (CK), "Dafugui" (DFG), and "Red Charm" (HSML) with accurate varieties (In lines 83-89).

 4. How did the authors decide on the HPLC condition of up to only 25% B solvent. In this way, most of the nonpolar compounds will not be eluted.

Reply: The gradient is not 25%; From 5% to 95%, gradient elution see line 109-110. We also have project experience in gradient conditions and have published several articles.

 5. Which part of plant did the authors extract? Sometimes deep freezing may change the tissues rapidly, affecting the constituents

Reply: The tissue site is the root of the P. lactiflora. Our method is freeze-dried. vacuum freeze dry technology referred to as freeze-drying technology, refers to the freezing of the drug solution at low temperature, and then sublimation drying in a vacuum environment to remove water, and then analytical drying after the sublimation, remove part of the combined water method. The technology prevents the physical, chemical and biological properties of the dried material to the greatest extent, and the damage to biological tissues and cell bodies is relatively small.

 6. How did authors select 70% methanol as the solvent? Why methanol and not ethanol? Was the condition of the sample kept equal to preserve the proanthocyanin contents?

Reply: Methanol has a simpler structure than ethanol, isopropyl alcohol and other small molecules, so it has stronger cell penetration and can dissolve a larger proportion of substances in cells, thus extracting more metabolites and higher metabolite content. Pure water can only extract polar substances in plants, such as glycosides, amino acids and other extremely polar substances, and the extraction efficiency is low. Pure methanol can extract some non-polar substances (such as free fatty acids, phospholipids) and moderate polar compounds (such as flavonoids phenolamines) in plants. 70% methanol water mixed solvent can extract compounds that can cover most of the metabolites in plants, and the extraction efficiency is higher. Therefore, the optimal method was finally determined to be 70% methanol.

 7. Please explain aim and methodology of volcano map, Venn plot, KEGG.

Reply:（1）The volcano map is drawn by ggplot2 (V3.3.0) in R language (V3.5.1); The Volcano Plot is mainly used to show the difference in the relative content of metabolites in the two varieties and the statistical significance of the difference (In line 146-148).

(2) The venn diagrams are drawn using the Venn Diagram (V1.6.20) package in R (V3.5.1). The function of the venn diagram is mainly to show the number of common and endemic substances between different comparison groups(In line 149,200-201). 

(3) KEGG enrichment: The process of annotating metabolites to the KEGG pathway according to the cpd_ID of differential metabolites, the pvalue of each pathway was calculated by hypergeometric test, and the ratio of the number of differentially expressed metabolites in the corresponding pathway to the total number of metabolites detected and annotated by this pathway was expressed by Rich Factor. (In line 150-154)

 8. How the statistical analysis was performed?

Reply: (1) PCA analysis: PCA used the built-in statistical prcomp function of R software (www.r-project.org/), and set prcomp function parameter scale=True, indicating that the data was normalized by unit variance scaling (UV).

(2) Hierarchical cluster analysis: The metabolite content data were normalized (Unit Variance Scaling, UV Scaling), and the heat map was drawn by R software Complex Heatmap package. The accumulation patterns of metabolites in different samples were analyzed by Hierarchical Cluster Analysis (HCA).

(3) After log2 transformation of the original data, OPLS-DA carries out Mean Centering processing, and then uses the Metabo AnalystR package OPLSR. Anal function in R software for analysis.

9. Supplementary information is not clear.

Reply: The name of the supplementary information has been changed to “Table Supplement” in “Supporting Information1”. The metabolite data of all samples were added in“Supporting Information2”, and the analysis data of metabolite differences among different varieties were added in “Supporting Information3”.

In addition, we intend to add a item has been added, please review and see if it can be added? 

Additional modification1

An author was added this time, Director Gao Kai was added mainly because he provided samples and gave identification as Director of "The National Peony and Peony Germplasm Bank". In addition, he modified the problems of the experimental design and participated in some modifications this time.

Concluding Remarks: 

The paper needs above mentioned corrections along with the English Language proof readings by some language expert. Then after fulfilling the corrections the paper will be acceptable. 

Reply: The above mentioned in the two reviewers have been revised and have also been polished by experts from Aiyide SCI polishing company. 

Sincerely,

Pro. Yonghui Li

College of Life Sciences

Luoyang Normal University 

Luoyang 

Henan 471934

P.R. China

---

## [Decision Letter · Decision Letter 1]

18 Dec 2023

PONE-D-23-30697R1Widely targeted metabolomics reveals differences in metabolites of Paeonia lactiflora cultivarsPLOS ONE

Dear Dr. Li,

Thank you for submitting your manuscript to PLOS ONE. After careful consideration, we feel that it has merit but does not fully meet PLOS ONE’s publication criteria as it currently stands. Therefore, we invite you to submit a revised version of the manuscript that addresses the points raised during the review process.

We look forward to receiving your revised manuscript.

Kind regards,

Sairah Hafeez Kamran, PhD

Academic Editor

PLOS ONE

Journal Requirements:

Reviewers' comments:

Reviewer's Responses to Questions

**Comments to the Author**

1. If the authors have adequately addressed your comments raised in a previous round of review and you feel that this manuscript is now acceptable for publication, you may indicate that here to bypass the “Comments to the Author” section, enter your conflict of interest statement in the “Confidential to Editor” section, and submit your "Accept" recommendation.

Reviewer #1: All comments have been addressed

Reviewer #2: (No Response)

2. Is the manuscript technically sound, and do the data support the conclusions?

Reviewer #1: Yes

Reviewer #2: Partly

3. Has the statistical analysis been performed appropriately and rigorously? 

Reviewer #1: Yes

Reviewer #2: Yes

4. Have the authors made all data underlying the findings in their manuscript fully available?

Reviewer #1: Yes

Reviewer #2: No

5. Is the manuscript presented in an intelligible fashion and written in standard English?

Reviewer #1: Yes

Reviewer #2: Yes

6. Review Comments to the Author

Reviewer #1: The manuscript entitled “Widely targeted metabolomics reveals differences in metabolites of Paeonia lactiflora cultivars” seems to be an interesting study in which authors made attempts to contribute the field. I have thoroughly reviewed the revised version of manuscript and it is concluded that all the suggestions have been addressed by authors.

Reviewer #2: Reviewer #2: Comments:

The authors study the differences in metabolites of Paeonia lactiflora cultivars which shows metabolomic analysis of the different cultivars. It is well written, but very brief and following comments should be addressed.

Authors replied and improved the manuscript, however, the replies are with poor language and syntax, even though manuscript has been improved. It may need further efforts to overall improve writing clarity and value of the work.

Major concerns:

1. Pure metabolomics lacks standards. It is recommended to confirm the major predicted components by classical HPLC quantification with the standards.

Reply: (1) High resolution mass spectrometry AB sciex TripleTOF6600 was used for qualitative detection of mixed samples for plant wide target, and then AB sciex6500 QTRAP was used for quantitative determination. Combined with the advantages of non-targeted and targeted metabolomics, high resolution mass spectrometry was used for accurate characterization. Triple quadrupole mass spectrometry with high sensitivity, high specificity and excellent quantitative capability is used as a complementary tool.

(2) The matching parameters in metabolite identification are Q1 precise molecular mass, secondary fragment, retention time, isotope.

(3) Qualitative and quantitative mass spectrometry analysis of metabolites in samples is based on the MetWare database (MWDB) and Multiple Reaction Monitoring (MRM). The identification of metabolites is based on the exact mass of metabolites, MS2 fragments, MS2 fragment isotope distribution and retention time (RT). Through the intelligent secondary spectrum matching method independently developed by MYWAY Company, the secondary spectrum and RT of metabolites in the project samples are intelligentially matched with the secondary spectrum and RT of the company database. The MS and MS2 tolerances are set to 20 ppm and the RT tolerances to 0.2min.

Thank you for clarification. Because the calibration curves are not shown and only relative abundance of ionized molecules can be shown using MS detector, it should be clearly stated in the manuscript. Still, the standards of major identified compounds should be purchased and analysed in line with the data from MS analysis to improve the chemical identification and quantification value of the work.

（4）The broad target detection method is to build a database of the standard product, and the metabolites without the standard product will be compared with the secondary spectrum in the public database or the literature. The source of substance identification at the time of database construction can be marked.

（5）Wide target quantitative mode: quantitative gold standard MRM mode, characteristic ion quantitative; Q3, RT, DP, CE of each substance will be optimized, and the peak shape will be better. Accuracy: AB sciex6500 QTRAP quantification; In the MRM model, the four-pole first screens the precursor ions (parent ions) of the target substance, and excludes the ions corresponding to other molecular weight substances to preliminarily eliminate the interference. After ionization induced by the collision chamber, the precursor ion breaks to form a lot of fragment ions, and then a characteristic fragment ion is selected through the triple four-pole filtration to eliminate the interference of non-target ions, so that the quantification is more accurate and the repeatability is better. After the metabolic mass spectrometry data of different samples were obtained, the peak area integral was performed for all the mass spectral peaks of substances, and the mass spectral peaks of the same metabolite in different samples were integral corrected (Fraga et al. 2010).

The language is hard to understand. The brief description could be added to the manuscript, thank you!

Fraga, C.G., et al., Signature-discovery approach for sample matching of a nerve-agent precursor using liquid chromatogd chemometrics. Anal Chem, 2010. 82(10): p. 4165-73.

Wide target methodology literature: Chen, W, Gong, L Guo, Z, et al,A Novel Integrated Method for Large-Scale Detection, Identification, and Quantification of Widely Targeted Metabolites: Application in the Study of Rice Metabolomics. Molecular Plant, 2013, 6(6): 1769-1780.

The relevant literature with high scores has been published：Zhao ML, Ren YJ , Wei W, Yang JM, Zhong QW and Zheng L. Metabolite Analysis of Jerusalem Artichoke(Helianthus tuberosus L.) Seedlings in Response to PolyethyleneGlycol-Simulated Drought Stress, Int.J. Mol. Sci. 2021, 22, 3294.

Please what means high scores?

2. Quantification of the components was not shown. Also, the identification confirmation based on a mother total ion and two distinct fragments was not shown.

Reply: Material content data, parent ions, and characteristic fragment ions are detailed in “Supporting Information2”: “ALL_sample_data.xlsx“. The data in the numerical table in the ALL_sample_data table uses the scientific counting method and is the relative content of metabolites, without units, by calculating the peak area formed by the characteristic ions of each substance in the detector; among Q1: Molecular weight of parent ions after the material is added by the electrospray ion source; Q3: Characteristic fragment ions; Sample column: Sample relative content (In lines 132-137).

Thank you. In the supporting information table, the data description is not clear, the form of data not easy to read. The precision for four decimals not seen. The explanation of each column missing. What is mix1 etc. It is unclear what Q1 and Q3 means. There sshould be two characteristics fragments in high resolution in order tomatch the identity of the compound. What * means. Overall, a minute amount of thousands of compounds with unclear quantity and quality gives low value for reader.

3. How did authors confirm the species varieties? Are there official names and genetic

identification available, could authors perform a genetic comparison study?

Reply: All the P. lactiflora samples were sourced from "The National Peony and Peony Germplasm Bank" (In September 2020, National Forestry and Grass Leaf Administration Announces List, http://chinahhxh.com/indexs.html ) at Luoyang Academy of Agriculture and Forestry Sciences. Provided by Gao Kai, Director of the Germplasm Resources Center, and identified by Professor Jin Mingxian from the Plant Classification Teaching and Research Office of Luoyang Normal University, they are classified as three varieties: "Fenyunu" (CK), "Dafugui" (DFG), and "Red Charm" (HSML) with accurate varieties (In lines 83-89).

The genetic confirmation and specimens information may be needed.

4. How did the authors decide on the HPLC condition of up to only 25% B solvent. In this way, most of the nonpolar compounds will not be eluted.

Reply: The gradient is not 25%; From 5% to 95%, gradient elution see line 109-110. We also have project experience in gradient conditions and have published several articles.

The response is not clear, “project experience”, there should be clear methodology in the article.

5. Which part of plant did the authors extract? Sometimes deep freezing may change the tissues rapidly, affecting the constituents

Reply: The tissue site is the root of the P. lactiflora. Our method is freeze-dried. vacuum freeze dry technology referred to as freeze-drying technology, refers to the freezing of the drug solution at low temperature, and then sublimation drying in a vacuum environment to remove water, and then analytical drying after the sublimation, remove part of the combined water method. The technology prevents the physical, chemical and biological properties of the dried material to the greatest extent, and the damage to biological tissues and cell bodies is relatively small.

OK, whether the plant material or its extract was freeze-drier should be clearly mentioned.

6. How did authors select 70% methanol as the solvent? Why methanol and not ethanol? Was the condition of the sample kept equal to preserve the proanthocyanin contents?

Reply: Methanol has a simpler structure than ethanol, isopropyl alcohol and other small molecules, so it has stronger cell penetration and can dissolve a larger proportion of substances in cells, thus extracting more metabolites and higher metabolite content. Pure water can only extract polar substances in plants, such as glycosides, amino acids and other extremely polar substances, and the extraction efficiency is low. Pure methanol can extract some non-polar substances (such as free fatty acids, phospholipids) and moderate polar compounds (such as flavonoids phenolamines) in plants. 70% methanol water mixed solvent can extract compounds that can cover most of the metabolites in plants, and the extraction efficiency is higher. Therefore, the optimal method was finally determined to be 70% methanol.

Thank you. Please add briefly to the manuscript.

7. Please explain aim and methodology of volcano map, Venn plot, KEGG.

Reply:（1）The volcano map is drawn by ggplot2 (V3.3.0) in R language (V3.5.1); The Volcano Plot is mainly used to show the difference in the relative content of metabolites in the two varieties and the statistical significance of the difference (In line 146-148).

(2) The venn diagrams are drawn using the Venn Diagram (V1.6.20) package in R (V3.5.1). The function of the venn diagram is mainly to show the number of common and endemic substances between different comparison groups (In line 149,200-201).

(3) KEGG enrichment: The process of annotating metabolites to the KEGG pathway according to the cpd_ID of differential metabolites, the pvalue of each pathway was calculated by hypergeometric test, and the ratio of the number of differentially expressed metabolites in the corresponding pathway to the total number of metabolites detected and annotated by this pathway was expressed by Rich Factor. (In line 150-154)

Please add the description to the manuscript.

8. How the statistical analysis was performed?

Reply: (1) PCA analysis: PCA used the built-in statistical prcomp function of R software (www.r-project.org/), and set prcomp function parameter scale=True, indicating that the data was normalized by unit variance scaling (UV).

(2) Hierarchical cluster analysis: The metabolite content data were normalized (Unit Variance Scaling, UV Scaling), and the heat map was drawn by R software Complex Heatmap package. The accumulation patterns of metabolites in different samples were analyzed by Hierarchical Cluster Analysis (HCA).

(3) After log2 transformation of the original data, OPLS-DA carries out Mean Centering processing, and then uses the Metabo AnalystR package OPLSR. Anal function in R software for analysis.

Please add to the manuscript. Please note that some of the description is improper and needs rephrasing: e.g.”Anal function in R software for analysis.”

9. Supplementary information is not clear.

Reply: The name of the supplementary information has been changed to “Table Supplement” in “Supporting Information1”. The metabolite data of all samples were added in“Supporting Information2”, and the analysis data of metabolite differences among different varieties were added in “Supporting Information3”.

Grammar of authors providing response is poor. It is recommended to clarify some parts such as below request whether appropriate.

In addition, we intend to add a item has been added, please review and see if it can be added?

Additional modification1

An author was added this time, Director Gao Kai was added mainly because he provided samples and gave identification as Director of "The National Peony and Peony Germplasm Bank". In addition, he modified the problems of the experimental design and participated in some modifications this time.

Concluding Remarks:

The paper needs above mentioned corrections along with the English Language proof readings by some language expert. Then after fulfilling the corrections the paper will be acceptable.

Reply: The above mentioned in the two reviewers have been revised and have also been polished by experts from Aiyide SCI polishing company.

Thank you very much!

7. PLOS authors have the option to publish the peer review history of their article (what does this mean?). If published, this will include your full peer review and any attached files.

Reviewer #1: **Yes: **Dr. Aamir Mushtaq

Reviewer #2: **Yes: **Michal Korinek

---

## [Author Response · Author response to Decision Letter 1]

12 Jan 2024

Response to Reviewers

[PONE-D-23-30697]

Dear Sir/Madam：

Thank you very much for your last letter and advice. This letter is in response to two reviewers questions.

Please note: Blue and red are the contents of this reply.

Major concerns:

1. Pure metabolomics lacks standards. It is recommended to confirm the major predicted components by classical HPLC quantification with the standards.

Reply: (1) High resolution mass spectrometry AB sciex TripleTOF6600 was used for qualitative detection of mixed samples for plant wide target, and then AB sciex6500 QTRAP was used for quantitative determination. Combined with the advantages of non-targeted and targeted metabolomics, high resolution mass spectrometry was used for accurate characterization. Triple quadrupole mass spectrometry with high sensitivity, high specificity and excellent quantitative capability is used as a complementary tool.

(2) The matching parameters in metabolite identification are Q1 precise molecular mass, secondary fragment, retention time, isotope.

(3) Qualitative and quantitative mass spectrometry analysis of metabolites in samples is based on the MetWare database (MWDB) and Multiple Reaction Monitoring (MRM). The identification of metabolites is based on the exact mass of metabolites, MS2 fragments, MS2 fragment isotope distribution and retention time (RT). Through the intelligent secondary spectrum matching method independently developed by MYWAY Company, the secondary spectrum and RT of metabolites in the project samples are intelligentially matched with the secondary spectrum and RT of the company database. The MS and MS2 tolerances are set to 20 ppm and the RT tolerances to 0.2min.

Thank you for clarification. Because the calibration curves are not shown and only relative abundance of ionized molecules can be shown using MS detector, it should be clearly stated in the manuscript. Still, the standards of major identified compounds should be purchased and analysed in line with the data from MS analysis to improve the chemical identification and quantification value of the work.

Reply: The wide target is a relative quantification method mainly used to compare and analyze the relative content changes of the same substance in different samples, without making calibration curves; The broad target has purchased standard products and established a database. The identification method has been marked, please refer to “Supporting Information3” (“ALL_sample _data.xlsx”). As shown below:

（4）The broad target detection method is to build a database of the standard product, and the metabolites without the standard product will be compared with the secondary spectrum in the public database or the literature. The source of substance identification at the time of database construction can be marked.

（5）Wide target quantitative mode: quantitative gold standard MRM mode, characteristic ion quantitative; Q3, RT, DP, CE of each substance will be optimized, and the peak shape will be better. Accuracy: AB sciex6500 QTRAP quantification; In the MRM model, the four-pole first screens the precursor ions (parent ions) of the target substance, and excludes the ions corresponding to other molecular weight substances to preliminarily eliminate the interference. After ionization induced by the collision chamber, the precursor ion breaks to form a lot of fragment ions, and then a characteristic fragment ion is selected through the triple four-pole filtration to eliminate the interference of non-target ions, so that the quantification is more accurate and the repeatability is better. After the metabolic mass spectrometry data of different samples were obtained, the peak area integral was performed for all the mass spectral peaks of substances, and the mass spectral peaks of the same metabolite in different samples were integral corrected (Fraga et al. 2010).

The language is hard to understand. The brief description could be added to the manuscript, thank you!

Reply: The brief description has been added to the manuscript in line 122-126.

Fraga, C.G., et al., Signature-discovery approach for sample matching of a nerve-agent precursor using liquid chromatogd chemometrics. Anal Chem, 2010. 82(10): p. 4165-73.

Wide target methodology literature: Chen, W, Gong, L Guo, Z, et al,A Novel Integrated Method for Large-Scale Detection, Identification, and Quantification of Widely Targeted Metabolites: Application in the Study of Rice Metabolomics. Molecular Plant, 2013, 6(6): 1769-1780.

The relevant literature with high scores has been published：Zhao ML, Ren YJ , Wei W, Yang JM, Zhong QW and Zheng L. Metabolite Analysis of Jerusalem Artichoke(Helianthus tuberosus L.) Seedlings in Response to Polyethylene Glycol-Simulated Drought Stress, Int. Mol. Sci. 2021, 22, 3294.

Please what means high scores?

Reply: The high score means that the published article is for Chinese Academy of Sciences Zone 2, with an impact factor of 5.6. It's a relatively high grade.

2. Quantification of the components was not shown. Also, the identification confirmation based on a mother total ion and two distinct fragments was not shown.

Reply: Material content data, parent ions, and characteristic fragment ions are detailed in “Supporting Information3”: “ALL_sample_data.xlsx”. The data in the numerical table in the ALL_sample_data table uses the scientific counting method and is the relative content of metabolites, without units, by calculating the peak area formed by the characteristic ions of each substance in the detector; among Q1: Molecular weight of parent ions after the material is added by the electrospray ion source; Q3: Characteristic fragment ions; Sample column: Sample relative content (In lines 132-137).

Thank you. In the supporting information table, the data description is not clear, the form of data not easy to read. The precision for four decimals not seen. The explanation of each column missing. What is mix1 etc. It is unclear what Q1 and Q3 means. There should be two characteristics fragments in high resolution in order to match the identity of the compound. What * means. Overall, a minute amount of thousands of compounds with unclear quantity and quality gives low value for reader.

Reply: “Supporting Information2”(ALL_sample_data.xlsx) has changed the previous scientific counting format to a regular format, with Q1 and Q3 having the precision for four decimals.

As shown below: 

Q1: Molecular weight of parent ion after the material is added with ions through the electric spray ion source; Q3: Characteristic fragment ions;

Mix sample is a mixed sample prepared by mixing all sample extracts. There are a total of 3 mix samples in this project, mainly used for quality control;

Metabolite Qualitative Analysis: High resolution mass spectrometry AB science TripleTOF6600 was used to qualitatively detect mixed samples for plant wide targeting, and high-resolution mass spectrometry was used for accurate qualitative analysis.

Metabolite identification is based on the precise mass of metabolites, MS2 fragments, isotopic distribution and retention time (RT) of MS2 fragments. Through Maiwei's self-developed intelligent secondary spectrum matching method, the secondary spectrum and RT of metabolites in project samples are intelligently matched with the company's database one by one. The MS and MS2 tolerance are set to 20 ppm, and the RT tolerance is 0.2 min.

Wide target methodology literature: Chen, W, Gong, L Guo, Z, et al,A Novel Integrated Method for Large-Scale Detection, Identification, and Quantification of Widely Targeted Metabolites: Application in the Study of Rice Metabolomics. Molecular Plant, 2013, 6(6): 1769-1780. 

3. How did authors confirm the species varieties? Are there official names and genetic

identification available, could authors perform a genetic comparison study?

Reply: All the P. lactiflora samples were sourced from "The National Peony and Peony Germplasm Bank" (In September 2020, National Forestry and Grass Leaf Administration Announces List, http://chinahhxh.com/indexs.html ) at Luoyang Academy of Agriculture and Forestry Sciences. Provided by Gao Kai, Director of the Germplasm Resources Center, and identified by Professor Jin Mingxian from the Plant Classification Teaching and Research Office of Luoyang Normal University, they are classified as three varieties: "Fenyunu" (CK), "Dafugui" (DFG), and "Red Charm" (HSML) with accurate varieties (In lines 83-89).

The genetic confirmation and specimens information may be needed.

Reply: Dear reviewers pro, the P. lactiflora samples were sourced from "The National Peony and Peony Germplasm Bank", and identified by Professor Jin. I think we should be able to confirm the species varieties. I have reviewed some literature and basically confirmed it this way. I don't know if it's possible? Thank you very much.

4. How did the authors decide on the HPLC condition of up to only 25% B solvent. In this way, most of the nonpolar compounds will not be eluted.

Reply: The gradient is not 25%; From 5% to 95%, gradient elution see line 109-110. We also have project experience in gradient conditions and have published several articles.

The response is not clear, “project experience”, there should be clear methodology in the article.

Reply: Clear methodology has been added to the article in line 110-112. 

5. Which part of plant did the authors extract? Sometimes deep freezing may change the tissues rapidly, affecting the constituents

Reply: The tissue site is the root of the P. lactiflora. Our method is freeze-dried. vacuum freeze dry technology referred to as freeze-drying technology, refers to the freezing of the drug solution at low temperature, and then sublimation drying in a vacuum environment to remove water, and then analytical drying after the sublimation, remove part of the combined water method. The technology prevents the physical, chemical and biological properties of the dried material to the greatest extent, and the damage to biological tissues and cell bodies is relatively small.

OK, whether the plant material or its extract was freeze-drier should be clearly mentioned.

Reply: The freeze-dried plant material has been clearly mentioned in line 97.

6. How did authors select 70% methanol as the solvent? Why methanol and not ethanol? Was the condition of the sample kept equal to preserve the proanthocyanin contents?

Reply: Methanol has a simpler structure than ethanol, isopropyl alcohol and other small molecules, so it has stronger cell penetration and can dissolve a larger proportion of substances in cells, thus extracting more metabolites and higher metabolite content. Pure water can only extract polar substances in plants, such as glycosides, amino acids and other extremely polar substances, and the extraction efficiency is low. Pure methanol can extract some non-polar substances (such as free fatty acids, phospholipids) and moderate polar compounds (such as flavonoids phenolamines) in plants. 70% methanol water mixed solvent can extract compounds that can cover most of the metabolites in plants, and the extraction efficiency is higher. Therefore, the optimal method was finally determined to be 70% methanol.

Thank you. Please add briefly to the manuscript.

Reply: The briefly content has been added to the manuscript in line 98-100.

7. Please explain aim and methodology of volcano map, Venn plot, KEGG.

Reply:（1）The volcano map is drawn by ggplot2 (V3.3.0) in R language (V3.5.1); The Volcano Plot is mainly used to show the difference in the relative content of metabolites in the two varieties and the statistical significance of the difference (In line 146-148).

(2) The venn diagrams are drawn using the Venn Diagram (V1.6.20) package in R (V3.5.1). The function of the venn diagram is mainly to show the number of common and endemic substances between different comparison groups (In line 149,200-201).

(3) KEGG enrichment: The process of annotating metabolites to the KEGG pathway according to the cpd_ID of differential metabolites, the pvalue of each pathway was calculated by hypergeometric test, and the ratio of the number of differentially expressed metabolites in the corresponding pathway to the total number of metabolites detected and annotated by this pathway was expressed by Rich Factor. (In line 150-154)

Please add the description to the manuscript.

Reply: Thank you very much. 

The methodology of volcano map has been added to the manuscript in line 155-157. 

The methodology of venn diagrams has been added to the manuscript in line 157-158 and 208-209. 

The methodology of KEGG enrichment has been added to the manuscript in line 159-163. 

8. How the statistical analysis was performed?

Reply: (1) PCA analysis: PCA used the built-in statistical prcomp function of R software (www.r-project.org/), and set prcomp function parameter scale=True, indicating that the data was normalized by unit variance scaling (UV). 

PCA uses the built-in statistical prcomp function in R software (www.r project. org/), and sets the prcomp function parameter scale=True to normalize the data using unit variance scaling (UV).

(2) Hierarchical cluster analysis: The metabolite content data were normalized (Unit Variance Scaling, UV Scaling), and the heat map was drawn by R software Complex Heatmap package. The accumulation patterns of metabolites in different samples were analyzed by Hierarchical Cluster Analysis (HCA).

Metabolite content data was normalized using Unit Variance Scaling, and a heatmap was drawn using the R software ComplexHetmap package to perform hierarchical cluster analysis (HCA) on the accumulation patterns of metabolites among different samples.

(3) After log2 transformation of the original data, OPLS-DA carries out Mean Centering processing, and then uses the Metabo AnalystR package OPLSR. Anal function in R software for analysis.

OPLS-DA analysis involved performing a log2 transformation on the raw data, followed by mean centering, and then using the OPLSR Anal function in the MetaboAnalystR package of R software for analysis.

Please add to the manuscript. Please note that some of the description is improper and needs rephrasing: e.g.” Anal function in R software for analysis.”

Reply: PCA analysis has been added to the manuscript in line 145-147.

HCA analysis has been added to the manuscript in line 147-149.

OPLS-DA analysis has been added to the manuscript in line 152-154.

9. Supplementary information is not clear.

Reply: The name of the supplementary information has been changed to “S1_raw_images” in “Supporting Information1” and “Table Supplement” in “Supporting Information2”. The metabolite data of all samples were added in “Supporting Information3”, and the analysis data of metabolite differences among different varieties were added in “Supporting Information4”.

---

## [Editor Report · Decision Letter 2]

22 Jan 2024

Widely targeted metabolomics reveals differences in metabolites of Paeonia lactiflora cultivars

PONE-D-23-30697R2

Dear Dr. Li,

We’re pleased to inform you that your manuscript has been judged scientifically suitable for publication and will be formally accepted for publication once it meets all outstanding technical requirements.

Kind regards,

Sairah Hafeez Kamran, PhD

Academic Editor

PLOS ONE
---

## [Editor Report · Acceptance letter]

21 Feb 2024

PONE-D-23-30697R2 

PLOS ONE

Dear Dr. Li, 

I'm pleased to inform you that your manuscript has been deemed suitable for publication in PLOS ONE. Congratulations! Your manuscript is now being handed over to our production team.

Kind regards, 

on behalf of

Dr. Sairah Hafeez Kamran 

Academic Editor

PLOS ONE